# Altered Phenotypes of Breast Epithelial × Breast Cancer Hybrids after ZEB1 Knock-Out

**DOI:** 10.3390/ijms242417310

**Published:** 2023-12-09

**Authors:** Alexander Merckens, Mareike Sieler, Silvia Keil, Thomas Dittmar

**Affiliations:** Institute of Immunology, Center for Biomedical Education and Research (ZBAF), Witten/Herdecke University, Stockumer Str. 10, 58448 Witten, Germany; alexander.merckens@uni-wh.de (A.M.); mareike.sieler@uni-wh.de (M.S.); silvia.keil@uni-wh.de (S.K.)

**Keywords:** cell–cell fusion, breast cancer, hybrid/ mixed E/M phenotype, ZEB1

## Abstract

ZEB1 plays a pivotal role in epithelial-to-mesenchymal transition (EMT), (cancer) cell stemness and cancer therapy resistance. The M13HS tumor hybrids, which were derived from spontaneous fusion events between the M13SV1-EGFP-Neo breast epithelial cells and HS578T-Hyg breast cancer cells, express ZEB1 and exhibit prospective cancer stem cell properties. To explore a possible correlation between the ZEB1 and stemness/ EMT-related properties in M13HS tumor hybrids, ZEB1 was knocked-out by CRISPR/Cas9. Colony formation, mammosphere formation, cell migration, invasion assays, flow cytometry and Western blot analyses were performed for the characterization of ZEB1 knock-out cells. The ZEB1 knock-out in M13HS tumor cells was not correlated with the down-regulation of the EMT-related markers N-CADHERIN (CDH2) and VIMENTIN and up-regulation of miR-200c-3p. Nonetheless, both the colony formation and mammosphere formation capacities of the M13HS ZEB1 knock-out cells were markedly reduced. Interestingly, the M13HS-2 ZEB1-KO cells harbored a markedly higher fraction of ALDH1-positive cells. The Transwell/ Boyden chamber migration assay data indicated a reduced migratory activity of the M13HS ZEB1-knock-out tumor hybrids, whereas in scratch/ wound-healing assays only the M13SH-8 ZEB1-knock-out cells possessed a reduced locomotory activity. Similarly, only the M13HS-8 ZEB1-knock-out tumor hybrids showed a reduced invasion capacity. Although the ZEB1 knock-out resulted in only moderate phenotypic changes, our data support the role of ZEB1 in EMT and stemness.

## 1. Introduction

The zinc finger E-box binding homeobox 1 (ZEB1) is the core epithelial-to-mesenchymal transition (EMT) transcription factor, which is associated with cell plasticity, proliferation, invasion and phenotypic transformation in distant sites [1,2,3]. Furthermore, high ZEB1 expression levels are commonly associated with poor prognosis for malignant tumors [1,2,3], which further highlights the important role of this transcription factor in cancer progression and metastasis.

The expression of ZEB1 in (cancer) cells is regulated by several positive and negative signaling pathways and regulatory networks [1,3]. Several signal transduction pathways have been identified that induce ZEB1 expression, such as transforming growth factor-β (TGF-β), Wnt, Ras-Raf-MAPK, PI3K/AKT, Notch and NF-κB signaling [1,3]. TGF-β is a well-known inducer of EMT and ZEB1 expression is mediated via TGF-β activated TGF-β receptor-SMAD signaling [4,5]. Likewise, the Ras-RAF-ERK and Wnt/β-catenin signaling were further confirmed as inducers of ZEB1 expression [6,7]. Negative regulators of ZEB1 expression are members of the microRNA-200 family, such as miR-200c and epithelial splicing regulatory protein 1 (ESRP1), hyaluronic acid synthase 2 (HAS2) and CD44 [8,9,10,11,12]. In this regard, the interplay of “miRNA-200c-3p-ZEB1” together with the “miRNA-34a-5p-SNAIL” reciprocal feedback loop has been suggested as core EMT network [5,12].

Due to its role in down-regulating E-CADHERIN (CDH1) expression, ZEB1 is commonly known as a driver of EMT, cancer progression and metastasis formation. Indeed, elevated ZEB1 levels have been found in several human cancers, such as lung cancer [13], colon cancer [14], glioblastoma [4] and breast cancer [15], which were further associated with advanced disease progression and higher metastatic spreading. However, a few studies have been published indicating that ZEB1 was rather not associated with EMT induction. For instance, data from Jägle and colleagues revealed that ZEB1 was neither sufficient nor required for EMT in LS174T colorectal cancer cells [16]. Here, the ectopic ZEB1 expression had only minor effects on cell morphology and invasive growth in three-dimensional spheroid cultures. In agreement with this, the expression of ZEB1 did not lead to repression of epithelial marker genes, and mesenchymal markers were not up-regulated by ZEB1. Moreover, the CRISPR/Cas9-mediated knock-out of ZEB1 did not affect the ability of ectopically expressed Snail1 to trigger a complete EMT in ZEB1-deficient LS174T cells [16]. Interestingly, recent findings of Sánchez-Tillo et al. demonstrated that ZEB1 had opposite functions in *KRAS-* and *BRAF*-mutant colorectal carcinomas [17]. While ZEB1 was correlated with a worse prognosis and a higher number of larger and undifferentiated mesenchymal *KRASG12D*-colorectal carcinomas, it was associated with a better prognosis and fewer, smaller and more differentiated *BRAFV600E* primary colorectal tumors [17]. Thus, ZEB1 can function as a tumor suppressor in *BRAF*-mutant colorectal cancer, which highlights the importance of analyzing the *KRAS*/*BRAF* mutational background in this disease.

Besides its role in EMT, ZEB1 may also promote (cancer) cell stemness and cancer therapy resistance [1,3,5,18,19,20]. Regarding EMT, it is commonly assumed, that cancer cells either remain in a non-motile epithelial (E) state or a motile mesenchymal (M) state [5,21]. However, an increasing body of evidence indicates, that cancer cells could also reside in a stable intermediate, or so-called mixed/hybrid epithelial–mesenchymal (E/M) state [12,18,19,22,23,24,25]. Thus, cancer cells exhibit some kind of phenotypic plasticity which allows them to switch between the E, E/M and M states. Mathematical modeling suggests that this mixed/hybrid E/M state and, hence, cancer cell plasticity is chiefly regulated by the “miRNA-200c-3p-ZEB1” reciprocal feedback loop [12,18,25]. Interestingly, cancer cells in the mixed/hybrid E/M state exhibited prospective cancer stem cells (CSCs) properties [19,26,27] suggesting that ZEB1 might be a determinant of stemness characteristics in tumor cells. Indeed, the depletion of ZEB1 suppressed stemness, colonization capacity, and in particular phenotypic/ metabolic plasticity of pancreatic cancer cells [28]. Similarly, the up-regulation of neurogenin 3, which is usually repressed by ZEB1, attenuated ZEB1-induced cancer stemness and symmetric CSC division [27]. Moreover, the single-cell sequencing of bevacizumab-resistant patient glioblastomas confirmed up-regulated mesenchymal genes, particularly glycoprotein YKL-40 and transcription factor ZEB1, in later clones, implicating these changes as treatment-induced. The CRISPR/Cas9-mediated KO and pharmacologic targeting of ZEB1 with honokiol reversed the mesenchymal gene expression and associated stem cell, invasion and metabolic changes of glioblastoma cells [20]. Thus, ZEB1 might be a target for preventing glioblastoma resistance [20].

However, data from Kröger and colleagues revealed that high levels of ZEB1 caused cells to complete an entire EMT and drove them into the M state which was incompatible with efficient tumor-initiating abilities [19]. The knockout of the ZEB1 gene together with forced expression of either SNAIL, SLUG, or TWIST caused cells to advance from an E state to the mixed/hybrid E/M state, which yielded cells that were about 38-fold more tumorigenic than cells in the E state [19]. Preca and colleagues reported a self-enforcing CD44s/ZEB1 feedback loop that maintains EMT and stemness properties in breast and pancreatic cancer cells [11]. Thereby, ZEB1 controls CD44s splicing by repression of ESRP1 in breast and pancreatic cancer, whereby CD44s itself activates ZEB1 expression [11]. Cancer cells with an active CD44s-ZEB1 regulatory loop exhibited an increased tumor-sphere initiation capacity suggesting that the CD44s-ZEB1 interplay renders tumor cell stemness independent of external stimuli [11]. These findings likely point to a more active role for ZEB1 in maintaining stemness in cancer cells even though neither SNAIL nor a mixed/hybrid E/M state was investigated in this study [11].

To gain CSC properties, a possible mechanism is cell-cell fusion of cancer cells with other cells. Cell–cell fusion itself represents a biological phenomenon that is mandatory for several physiological processes, such as fertilization, placentation, myogenesis, osteoclastogenesis, and wound healing/ tissue regeneration [29,30,31,32,33,34,35,36,37]. Additionally, cell–cell fusion takes also place in pathophysiological conditions, such as infection of host cells with enveloped viruses and cancer [38,39,40,41,42,43,44,45]. With regard to cancer, it is known that cancer cells could fuse with normal cells, such as macrophages, fibroblasts, and stem cells, thereby giving rise to tumor hybrids possessing novel properties, such as immune escape, an increased drug resistance, an enhanced metastatic capacity, and prospective CSC properties [38,39,40,41,42,43,44,45,46,47,48,49]. Albeit both the spontaneous fusion frequencies of cancer cells/ normal cells and the overall survival rates of tumor hybrids appear to be rather low (up to 1% each), mathematical modeling revealed that fusion-mediated recombination can have a profound impact on accelerated diversification of tumor cell populations and an increasing intratumoral heterogeneity [50].

In a previous study, we demonstrated that the M13HS tumor hybrids, which were derived from spontaneous fusion events between human M13SV1-EGFP-Neo breast epithelial cells and human HS578T-Hyg breast cancer cells [51] exhibited prospective CSC properties [52]. Interestingly, the M13HS tumor hybrids revealed a co-expression of the EMT transcription factors ZEB1 and SNAIL, possessed a higher fraction of ALDH+ positive cells, formed larger mammospheres and migration was induced by EGF [52]. Similarly, the M13HS-2 and -8 tumor hybrids, but not parental M13SV1-EGFP-Neo breast epithelial cells and HS578T-Hyg breast cancer cells, responded to the chemokine CCL21 with an increased locomotory activity [53]. Thus, these two tumor hybrids together with parental HS578T-Hyg breast cancer cells were chosen to explore the role of ZEB1 and its CRISPR/Cas9 KO in migration, invasion and prospective CSC properties.

## 2. Results

### 2.1. Succesful ZEB1-KO in HS578T-Hyg and M13HS-2 and -8 Tumor Hybrids

We have demonstrated in a previous study that the M13HS-2 and -8 tumor hybrids co-expressed SNAIL and ZEB1, while parental human M13SV1-EGFP-Neo breast epithelial cells were only positive for SNAIL and human HS578T-Hyg breast cancer cells only expressed ZEB1 [52]. Because SNAIL and particularly ZEB1 have been proposed as markers for the mixed/hybrid E/M phenotype [18,19,54], ZEB1 was specifically knocked-out by CRISPR/Cas9. Data are summarized in Figure 1 and clearly show a stable ZEB1-KO in HS578T-Hyg, M13HS-2 and M13HS-8 cells. Since M13SV1-EGFP-Neo cells lack ZEB1 expression, no CRISPR/Cas9 ZEB1 KO was performed.

### 2.2. ZEB1-KO Is Not Correlated to a Markedly Altered E/M Gene Expression Profile

Western blot studies were performed to investigate whether ZEB1-KO was associated with an altered protein expression of M-genes and up-regulation of E-genes. CDH1 and CYTOKERATIN-5 (CK5) were chosen for E-genes and N-CADHERIN (CDH2), VIMENTIN (VIM), and WNT5A for M-genes (Figure 2). Furthermore, the expression level of the EMT transcription factor SNAIL was investigated.

Interestingly, the protein expression profile of E- and M-genes in ZEB1-KO cells was rather comparable to wildtype cells (Figure 2), which was unexpected. ZEB1 is a well-known repressor of CDH1 expression [55,56] and therefore the re-induction of this adhesion molecule was assumed. Interestingly, higher SNAIL expression levels were found in all ZEB1-KO cells. Thereby, the M13HS-2 and -8 ZEB1-KO tumor hybrids showed markedly increased SNAIL expression levels, whereas rather weak SNAIL expression levels were observed in HS578T-Hyg ZEB1-KO cells (Figure 2). Expression of the active form of WNT5A was only detected in HS578T-Hyg breast cancer cells. Interestingly, a weak WNT5A up-regulation was observed in the M13HS-2 and -8 ZEB1-KO tumor hybrids (Figure 2).

### 2.3. ZEB1-KO Cells Exhibit a More Round-Shaped Epithelial Morphology

Confocal laser scanning microscopy studies were performed to analyze whether ZEB1-KO was associated with an altered cellular morphology. As expected, the M13SV1-EGFP-Neo breast epithelial cells exhibited a round-shaped epithelial morphology, whereas the HS578T-Hyg breast cancer cells and M13HS-2 and -8 tumor hybrids showed a spindle-like, elongated mesenchymal phenotype (Figure 3).

Interestingly, the ZEB1-KO cells possessed a more round-shaped epithelial-like phenotype (Figure 3).

### 2.4. miRNA-34a-5p, but Not miRNA-200c-3p Levels Are Slightly Increased in ZEB1-KO Cells

The interplay of “miRNA-34a-5p-SNAIL” and “miRNA-200c-3p-ZEB1” has been suggested as core EMT networks [5,12]. Since miRNA-200c-3p represses ZEB1 expression and vice versa [5,8,12], the relative miRNA’s expression levels were determined by qPCR (Figure 4A,B).

Significantly lower miRNA-34a-5p and miRNA-200c-3p expression levels were observed in the HS578T-Hyg breast cancer cells, M13HS-2 and -8 tumor hybrids and their ZEB1-KO variants, in comparison to the M13SV1-EGFP-Neo breast epithelial cells. However, miRNA data were only partially congruent to SNAIL and ZEB1 protein expression data.

MiRNA-34a-5p was clearly detectable in the M13SV1-EGFP-Neo breast epithelial cells, which remains ambiguous due to the cells’ marked SNAIL expression levels (Figure 2 and Figure 4A). Likewise, no, or even low to moderate SNAIL expression levels were observed in the HS578T-Hyg breast cancer cells and M13HS-2 and -8 tumor hybrids (Figure 2) despite significantly low miRNA-34a-5p expression levels (Figure 4A). ZEB1-KO resulted only in moderately altered miRNA-34a-5p levels (Figure 4A). In comparison to the HS578T-Hyg breast cancer cells, the miRNA-34a-5p levels were about 2-fold lower in the HS578T-Hyg ZEB1-KO cells (Figure 4A). Whether this was correlated to the observed slightly increased SNAIL expression in these cells (Figure 2) is not yet clear. In contrast, miRNA-34a-5p expression levels were weakly higher in M13HS-2 and -8 ZEB1-KO cells than in wildtype cells. However, as shown in Figure 2, the SNAIL expression was markedly higher in the M13HS-2 and -8 ZEB1-KO cells than in M13HS-2 and -8 wildtype cells.

The miRNA-200c-3p data correlated well with wildtype cells. The HS578T-Hyg breast cancer cells and M13HS-2 and -8 tumor hybrids were ZEB1 positive, which was in line with low levels of the miRNA-200c-3p in these cells (Figure 4B). Similarly, the M13SV1-EGFP-Neo breast epithelial cells lacked ZEB1 expression due to the high miRNA-200c-3p levels (Figure 2 and Figure 4B). However, the finding, that the miRNA-200c-3p levels were only very weakly altered in the ZEB1-KO cells was unexpected. Given that ZEB1 is a repressor of miRNA-200c-3p [5,12], one would have expected much higher miRNA-200c-3p levels in the ZEB1-KO variants (Figure 4B). However, the miRNA-200c-3p expression was only 2.6-fold (M13HS-8 ZEB1-KO) to 3.6-fold (M13HS-2 ZEB1-KO) increased in the ZEB1-KO variants.

### 2.5. ZEB2 Levels of ZEB1-KO Cells Were Comparable to Wildtype Cells

Like ZEB1, ZEB2 is also a master regulator of EMT and CDH1 repression [57,58], and its expression is related to miRNA-200c-3p and vice versa [59,60]. Thus, Western blot studies were performed to investigate the ZEB2 expression level in wildtype cells and ZEB1-KO cells (Figure 5).

ZEB2 expression was detected in all cell lines, which agrees with previously published data [52]. Indeed, the HS578T-Hyg ZEB1-KO cells revealed a markedly higher ZEB2 expression than the HS578T-Hyg breast cancer cells. By contrast, comparable ZEB2 expression levels were observed in the M13HS-2 and -8 tumor hybrids and their ZEB1-KO variants. Whether this might be an explanation for still low miRNA-200c-3p expression levels in the ZEB1-KO variants is unclear, since Western blots also show ZEB2 expression in the miRNA-200c-3p positive M13SV1-EGFP-Neo breast epithelial cells (Figure 4B and Figure 5).

### 2.6. CD44/CD104 Expression Profile

CD44 and CD104 have been suggested as markers for discrimination of the E, mixed/hybrid E/M and M state of cancer cells [19,23]. Thus, the ZEB1-KO cells and wildtype cells were analyzed for CD44/CD104 expression by flow cytometry (Figure 6).

The flow cytometry data indicated two CD44/CD104 populations in all the analyzed cell lines. The major population was comprised CD44^+^/CD104^-^ cells (R2), whereas in the second and much smaller population (R3), a faint CD104 expression was observed (Figure 6). The R3 population was much higher in M13HS-2 (19.21 ± 0.42%) and M13HS-8 (18.69 ± 0.52%) tumor hybrids than in the M13SV1-EGFP-Neo breast epithelial cells (8.41 ± 2.94%) and HS578T-Hyg breast cancer cells (3.58 ± 1.32%) (Figure 6). Interestingly, the R3 population was diminished in all the ZEB1-KO cells, whereby the highest reduction was observed in the M13HS-2 ZEB1-KO cells (10.98 ± 4.02% vs. 19.21 ± 0.42% (M13HS-2)). In contrast, the R3 population was only slightly diminished in the HS578T-Hyg ZEB1-KO and M13HS-8 ZEB1-KO cells.

### 2.7. M13HS-2 ZEB1-KO Cells Exhibit a Markedly Enriched ALDH1 Positive Population

In addition to the CD44/CD104 expression pattern, we analyzed the cells for ALDH1 expression, which has been identified as a marker of normal and malignant human mammary stem cells [61]. Interestingly, while the fractions of the ALDH1-positive cells were comparable between the HS578T-Hyg breast cancer cells, the M13HS-8 tumor hybrids and their ZEB1-KO variants, the ALDH1-positive cells (27.3 ± 4.0%; Figure 7) were highly enriched in the M13HS-2 ZEB1-KO cells.

### 2.8. ZEB1-KO Cells Exhibit a Decreased Colony Formation Capacity

As ZEB1 has been proposed as a determinant of stemness, the colony formation capacity of ZEB1-KO variants was analyzed. In fact, the ZEB1-KO cells exhibited a significantly decreased colony formation capacity in comparison to their wildtype counterparts (Figure 8).

Thereby, the highest decrease was observed in the M13HS-2 ZEB1-KO tumor hybrids (about two-thirds less compared to M13HS-2 cells), whereas only a moderate reduction was found in the M13HS-8 ZEB1-KO cells (about one-third less than of M13HS-8 cells (Figure 8). The colony formation capacity of the HS578T-Hyg ZEB1-KO breast cancer cells was about 50% lower than that of the HS578T-Hyg wildtype breast cancer cells (Figure 8).

### 2.9. M13HS ZEB1-KO Cells Exhibit a Decreased Mammopshere Formation Capacity

Next, the mammosphere formation capacity of the cells was investigated. In accordance with previous data [52,62], both the M13HS-2 and -8 tumor hybrids exhibited an increased mammosphere formation capacity as compared to the parental M13SV1-EGFP-Neo breast epithelial cells and HS578T-Hyg breast cancer cells (Figure 9).

Interestingly, more mammospheres were derived from the HS578T-Hyg ZEB1-KO cells as compared to the HS578T-Hyg cells (HS578T-Hyg: 1.17 ± 0.29 vs. HS578T-Hyg ZEB1-KO: 6.04 ± 0.86; Figure 9). Whether this was attributed to slightly increased SNAIL expression levels in these cells (Figure 2) remains unclear. In contrast, the mammosphere formation capacity of the M13HS-2 ZEB1-KO and -8 ZEB1-KO tumor hybrids was markedly and significantly reduced (Figure 9). While the M13HS-2 ZEB1-KO still exhibited some mammosphere formation capacity, virtually no mammospheres were derived from the M13HS-8 ZEB1-KO cells (Figure 9).

### 2.10. ZEB1-KO Cells Exhibit Different Migratory Properties

To analyze the migratory properties of the ZEB1-KO cells we first performed a scratch/ wound-healing assay. Data are summarized in Figure 10 and clearly show, that each ZEB1-KO variant possesses a unique migratory behavior.

For instance, the scratch/ wound was much faster closed by HS578T-Hyg ZEB1-KO cells than by HS578T-Hyg suggesting, that the ZEB1-KO variant exhibited a higher migratory capacity. In contrast, the migratory behavior of the M13HS-2 cells and M13HS-2 ZEB1-KO cells was rather comparable, whereas the migratory activity of the M13HS-8 ZEB1-KO cells was significantly diminished in comparison to M13HS-8 cells (Figure 10).

### 2.11. ZEB1-KO Cells Exhibit a Decrased Migratory Activity in Transwell/ Boyden Chamber Assays

In addition to the scratch/ wound-healing assay, a Transwell/ Boyden chamber assay was performed to study the chemotactic behavior of the cells. Here, all the ZEB1-KO variants exhibited a markedly and partially significant decreased migratory activity (Figure 11).

The migratory activity of the HS578T-Hyg ZEB1-KO cells was rather slightly decreased as compared to the HS578T-Hyg breast cancer cells (HS578T-Hyg: 230 ± 12 cells vs. HS578T-Hyg ZEB1-KO: 158 ± 7 cells). In contrast, the migratory activity of the M13HS-2 ZEB1-KO and -8 ZEB1-KO variants was virtually completely abrogated in comparison to the M13HS-2 and -8 tumor hybrids (Figure 11).

### 2.12. ZEB1-KO Cells Exhibit Different Invasion Capacities

Finally, the invasion capacity of the cells was analyzed. Data are summarized in Figure 12 and show, that the M13SV1-EGFP-Neo breast epithelial cells and HS578T-Hyg breast cancer cells exhibited no invasive capacities. In contrast, both the M13HS-2 and -8 tumor hybrids possessed significantly enhanced invasive capacities (Figure 12).

In contrast to normal Transwell/ Boyden chamber assay, we only observed a reduced invasive capacity for the M13HS-8 ZEB1-KO cells, whereas the invasion capacity of the HS578T-Hyg ZEB1-KO and M13HS-2 ZEB1-KO cells was similar to their wildtype counterparts (Figure 12).

## 3. Discussion

In addition to its critical role in the EMT process, the transcription factor ZEB1 might also be a determinant of (cancer) cell stemness and cancer therapy resistance [1,3,5,18,19]. To explore its function in the M13HS tumor hybrids, which were derived from spontaneous fusion events between the human M13SV1-EGFP-Neo breast epithelial cells and the human HS578T-Hyg breast cancer cells [51], ZEB1 was knocked-out by CRISPR/Cas9. In this work, the validated ZEB1 gRNA sequence of Kröger et al. was used [19], which gave rise to a stable ZEB1-KO in the HS578T-Hyg breast cancer cells and M13HS-2 and -8 tumor hybrids.

Kröger et al. reported about three distinct populations (E, mixed E/M, and M) that were derived from HMLER human mammary epithelial cells, through multiple successive cycles of flow cytometry sorting [19]. These subpopulations expressed a specific, so-called E- and M-gene expression pattern and were further distinguishable from each other by a differential CD44/CD104 expression pattern [19]. Moreover, the so-called mixed E/M phenotype exhibited prospective CSC properties [19]. SNAIL and ZEB1 have been suggested as determinants of these three states, whereby cells in an E state are SNAIL and ZEB1 negative, cells in a mixed/hybrid-E/M state co-express SNAIL and ZEB1 and cells in an M state express high levels of ZEB1, but low levels of SNAIL [19].

Comparison of the E- and M-gene expression pattern of the M13SV1-EGFP-Neo breast epithelial cells, HS578T-Hyg breast cancer cells, and M13HS-2 and -8 tumor hybrids with the respective protein expression profile of E-, mixed E/M and M HMLER subpopulations revealed, that the M13SV1-EGFP-Neo cells exhibited a “classical” E phenotype. The cells express CDH1 and CK5, but lack CDH2, VIM and ZEB1. In contrast, the HS578T-Hyg breast cancer cells are in a M state due to the expression of “classical” M markers, such as CDH2, VIM, ZEB1 and WNT5A [19]. Similarly, neither SNAIL nor CDH1 were expressed, which is consistent with the data of Kröger and colleagues [19]. M13HS-2 and -8 tumor hybrids co-expressed ZEB1 and SNAIL, whereby the SNAIL expression levels were higher in the M13HS-2 than in M13HS-8 cells. Whether this co-expression of SNAIL and ZEB1 may point to a prospective mixed/hybrid E/M phenotype of the tumor hybrids is not yet clear. On the one hand, the tumor hybrids expressed “classical” M-genes, such as CDH2 and VIM, suggesting that they are rather in a M state. On the other hand, the M13HS-2 and -8 tumor hybrids lack expression of the active form of WNT5A, which has been suggested as a M state marker [19]. The CD44/CD104 expression marker analysis revealed that most of the HS578T-Hyg breast cancer cells were CD44^+^/CD104^-^, which is further in agreement with a M state. In contrast, clear CD44^+^CD104^low^ subpopulations were observed in the M13HS-2 and -8 tumor hybrids. Whether these subpopulations represent tumor hybrids in a mixed/hybrid E/M state is not yet clear. On the one hand, CD104 has been suggested as a marker for the mixed/hybrid E/M state and prospective CSCs [19,23]. On the other hand, the CD44^+^CD104^low^ subpopulation was also detected in the M13SV1-EGFP-Neo breast epithelial cells. The HMLER cells in an E-state expressed high levels of CD104, but low levels of CD44 [19], which is contrasting to the CD44^+^/CD104^low^ expression pattern of M13SV1-EGFP-Neo cells. The M13HS-2 and -8 tumor hybrids possessed an enhanced colony formation and mammosphere formation capacity, which would agree with a prospective mixed/hybrid-E/M phenotype and CSC properties. However, in this study, the colony formation capacity of the HS578T-Hyg cells was comparable to M13HS-2 and -8 tumor hybrids, despite a M state and a lower fraction of CD44^+^CD104^low^ cells. Similarly, about 10% of the M13SV1-EGFP-Neo cells were CD44^+^CD104^low^, but the cells exhibited no mammosphere formation capacity, which is opposite to the assumption that CD104 is a prospective CIC marker. Thus, further research is necessary to investigate whether the CD44^+^CD104^low^ subpopulation of the M13HS-2 and -8 tumor hybrids exhibit prospective CSC properties.

ZEB1-KO resulted in rather unexpected results, since both miRNA-200c-3p levels and CDH1 expression levels remained unchanged in the HS578T-Hyg ZEB1-KO breast cancer cells and M13HS ZEB1-KO tumor hybrids. The microRNA-200 family is the main antagonist of ZEB1 expression and vice versa [8,9]. For instance, stable knock-down of ZEB1 expression in the MDA-MB-231 breast cancer cells with specific ZEB1 shRNA, resulted in markedly elevated (about 800-fold) miRNA-200c expression levels [9]. Similarly, significantly decreased ZEB1 mRNA levels were determined in the MDA-MB-231 cells after transient overexpression of miRNA-200c [9]. Thus, the stable ZEB1-KO should have correlated with a marked up-regulation of miRNA-200c-3p expression in the cells. In fact, only moderately altered ΔC_T_ values of about 1.5 to 1.6 were observed in this work. In accordance therewith, the CDH1 expression levels also remained unaltered in ZEB1-KO variants in comparison to untreated wildtype cells. Both, Burk et al. and Sundarayan et al. demonstrated up-regulated CDH1 expression levels in MDA-MB-231 breast cancer cells after specific ZEB1 shRNA-mediated ZEB1 knock-down [8,9]. Even Kröger and colleagues observed a faint CDH1 up-regulation in M-ZEB1-KO cells [19].

The reasons for rather unaltered miRNA-200c-3p and CDH1 expression levels in the ZEB1-KO variants remain unclear. Sanger sequencing revealed one additional nucleotide in the ZEB1 gene of the M13HS-2 ZEB1-KO and -8 ZEB1-KO tumor hybrids, whereas gene editing was much more chaotic in HS578T-Hyg breast cancer cells (Figure 1A). Thus, ZEB1 expression in all the ZEB1-KO cells was most likely impaired by an early stop codon due to a frameshift.

One possibility for why CDH1 is still expressed in the ZEB1-KO variants might be related to the cell´s SNAIL expression levels. SNAIL is another EMT transcription factor and a known repressor of CDH1 [5,63,64]. Indeed, weakly to markedly enhanced SNAIL expression levels were observed in all ZEB1-KO variants. However, M13SV1-EGFP-Neo breast epithelial cells expressed CDH1 despite a marked SNAIL expression, which remains ambiguous. Jägle and colleagues demonstrated, that ZEB1 was neither sufficient nor required for EMT in the LS174T colorectal cancer cells [16]. Epithelial marker genes were not repressed and mesenchymal markers were not upregulated by ectopic expression of ZEB1. Similarly, the ability of ectopically expressed SNAIL to trigger a complete EMT was not affected in ZEB1 CRISPR/Cas9-KO LS174T colorectal cancer cells [16]. These data tend to indicate that ZEB1 might be likely not involved in EMT. However, since many other data support the role of ZEB1 in EMT [1,2,3], it needs to be clarified whether the results obtained on LS174T colorectal cancer cells might be cell line specific.

It cannot be further ruled out, that unaltered miRNA-200c-3p and CDH1 expression levels in the ZEB1-KO variants were related to ZEB2 expression. Like ZEB1, ZEB2 is also a master regulator of EMT and CDH1 repression [57,58], and its expression is related to miRNA-200c-3p and vice versa [59]. Indeed, the HS578T-Hyg and M13HS tumor hybrids expressed ZEB2 [52], whereby markedly higher ZEB2 levels were observed in the HS578T-Hyg ZEB1-KO cells as compared to the HS578T-Hyg wildtype cells. This could be an explanation for the still low miRNA-200c-3p levels in the ZEB1-KO cells. However, ZEB2 is also expressed in the M13SV1-EGFP-Neo breast epithelial cells [52], which are positive for miRNA-200c-3p expression. Similarly, it remains unclear why shRNA-mediated knockdown of ZEB1 in the MDA-MB-231 breast cancer cells was correlated with elevated miRNA-200c-3p levels [9] although the MDA-MB-231 cells express ZEB2 too [65,66]. Thus, the prospective impact of ZEB2 in regulating the miRNA-200c-3p expression should be clarified in ongoing studies.

In addition to its role in EMT and cancer progression, an increasing body of evidence suggests that ZEB1 might also be a determinant of (cancer) cell stemness [1,3,5,18,19]. In this regard, studies of Zhou et al. demonstrated, that ZEB1 gain-of-function transfection in MDA-MB-231 breast cancer cells resulted in a higher tumorsphere formation capacity, a higher percentage of side-population cells, a higher fraction of CD44^+^CD24^-^ breast CSC population and an increased ALDH activity [27]. Similarly, overexpression of miR-199a-3p significantly reduced tumor growth, cell proliferation, sphere formation capacity and ALDH expression of A549 lung carcinoma cells, due to down-regulation of ZEB1 [67]. All these effects were inverted with overexpression of ZEB1 [67], which substantiates the prospective role of ZEB1 in (cancer) cell stemness. Likewise, the ALDH+ head and neck cancer cells also exhibited higher ZEB1 expression levels [68]. Interestingly, prospective CD44^+^CD117^+^CD133^+^ head and neck CSCs possessed lower ZEB1 and ALDH expression levels, but still exhibited CSC properties due to up-regulation of NANOG [68]. Briefly, all these findings support a correlation between ZEB1 and ALDH expression. However, the data presented here did not support this correlation. In fact, the HS578T-Hyg and HS578T-Hyg ZEB1-KO cells, as well as M13HS-8 and M13HS-8 ZEB1-KO cells possessed comparable amounts of ALDH+ cells. Moreover, the ZEB1-KO in M13HS-2 tumor hybrids resulted in a markedly increased (and reproducible) fraction of ALDH+ cells, which contrasts with the above-summarized correlation of ZEB1 and ALDH expression. However, different pathways have been identified that regulate ALDH1 expression and activity, such as WNT/β-catenin signaling, MUC1-C/ERK/C/EBPβ, retinoic acid and NOTCH signaling. Hence, it cannot be ruled out, that the markedly increased ALDH1 activity of M13HS-2 ZEB1-KO cells was related to one or more of these signaling pathways. For instance, M13HS-2 ZEB1-KO cells likely expressed higher levels of the active form of WNT5A. Whether this may indicate a prospective higher activity of WNT/β-catenin signaling in M13HS-2 ZEB1-KO remains to be elucidated in future studies.

Studies by Vandamme et al. revealed, that a reversible switching of the ZEB2/ZEB1 ratio may play a role in the proliferation, invasion and metastatic dissemination of melanoma cells [69]. Thereby, high ZEB2 and low ZEB1 expression levels were associated with primary tumor growth, differentiation, metastatic outgrowth and survival [69]. In contrast, melanoma cell invasion was driven by high ZEB1 and low ZEB2 expression levels [69]. As shown here, the parental M13SV1-EGFP-Neo breast epithelial cells and HS578T-Hyg breast cancer cells, as well as their M13HS tumor hybrids and the ZEB1-KO variants exhibit different colony and mammosphere formation capacities and migration/invasion properties. Whether these differences were related to differential ZEB2/ZEB1 ratios is rather unlikely. The colony and mammosphere formation capacities of the ZEB1-KO cells were markedly diminished as compared to wildtype cells, which is opposite to the findings of Vandamme and colleagues [69]. They showed, that a high ZEB2/ZEB1 ratio was associated with proliferation, whereas a low ratio favored invasion and migration [69]. Due to ZEB1-KO, the ZEB2/ZEB1 ratio must be higher in the ZEB1-KO cells and should therefore correlate with increased proliferation and colony formation, which was not observed here. In fact, comparable ZEB2 expression levels in the ZEB1-KO cells and wildtype cells were observed.

Similarly, the cell migration and invasion data of wildtype and ZEB1-KO cells remain ambiguous. Transwell/ Boyden chamber data indicate, that the migratory activity of ZEB1-KO was markedly diminished in comparison to wildtype cells, which is in agreement with the pivotal role of ZEB1 in EMT, cancer progression and metastasis formation [1,2,3]. However, scratch assay/ wound-healing studies yielded different results. Here, only the M13HS-8 ZEB1-KO cells possessed a decreased wound-healing capacity, whereas the scratch/ wound-healing capacity of the M13HS-2 and M13HS-2 ZEB1-KO cells was comparable. In contrast, the scratch/ wound was significantly faster closed by the HS578T-Hyg ZEB1-KO cells than by the HS578T-Hyg wildtype breast cancer cells. Due to its role in EMT and cell migration, the ZEB1-KO should have been rather associated with a diminished scratch/ wound closure of HS578T-Hyg ZEB1-KO and M13HS-2 ZEB1-KO cells. The reasons for these unexpected data are not yet clear. For instance, Zhao and colleagues demonstrated, that shRNA-mediated knock-down of ZEB1 was correlated with a reduced migration and invasion ability of B15F10 melanoma cells [70]. Similarly, the migration of human hepatocellular carcinoma cells was impaired in ZEB1 siRNA-treated cells [71]. Interestingly, the inducible knock-down of ZEB1 resulted in a partial EMT phenotype in PC-3 prostate cancer cells including co-expression of epithelial and mesenchymal markers, a mixed E/M morphology and an increased invasion and migration capacity [72]. Whether the findings of Kitz and colleagues might be an explanation for the observed higher migratory activity of HS578T-Hyg ZEB1-KO cells is rather unlikely. For instance, the authors observed a mixed E/M phenotype in the ZEB1 knock-down PC-3 cells including CDH1 up-regulation [72]. As mentioned above, no CDH1 up-regulation was observed in ZEB1-KO cells in this study. Similarly, we do not conclude from the Western blot data that HS578T-Hyg ZEB1-KO cells have acquired a mixed/hybrid E/M-state.

While invasion data of the M13HS-2 and -8 tumor hybrids suited well to the assumption, that tumor hybrids often exhibit novel properties, such as an enhanced metastatic capacity [30,38,39,73,74], it remains unclear why the invasive capacity of M13HS-2 ZEB1-KO cells was not diminished, but rather comparable to M13HS-2 wildtype cells. ZEB1 is a well-known inducer of EMT and high ZEB1 levels are associated with a higher invasive capacity of cancer cells, whereas invasion is impaired upon ZEB1 knock-down [4,75,76]. These data are in agreement with other findings of M13HS-2 ZEB1-KO tumor hybrids that do not behave as expected. At present, we do not have a suitable explanation for this observation.

## 4. Materials and Methods

### 4.1. Cell Culture

The M13SV1-EGFP-Neo cells were derived from the M13SV1 human breast epithelial cells (kind gift of James Trosko, Michigan State University, East Lansing, MI, USA [77]) and were stably transfected with the pEGFP-Neo plasmid [51]. The HS578T-Hyg human breast cancer cells were derived from HS578T cells (HTB 126; LGC Standards GmbH, Wesel, Germany) by stable transfection with the pKS-Hyg plasmid. M13HS-2 and -8 hybrid cells were derived from spontaneous fusion events between the M13SV1-EGFP-Neo cells and HS578T-Hyg cells [51,78]. All cells were maintained in RPMI 1640 media (PAN Biotech GmbH, Aidenbach, Germany) supplemented with 10% fetal calf serum (PAN Biotech GmbH, Aidenbach, Germany) and 100 U/mL penicillin/ 0.1 mg/mL streptomycin (PAN Biotech GmbH, Aidenbach, Germany). The following supplements were further added to the culture medium. M13SV1-EGFP-Neo: 10 µg/mL recombinant human epidermal growth factor (rhEGF), 5 µg/mL human recombinant insulin, 0.5 µg/mL hydrocortisone, 4 µg/mL human transferrin, 10 nM β-estrogen, and 400 µg/mL G418 (all supplements were purchased from Merck KGaA, Darmstadt, Germany). HS578T-Hyg: 200 µg/mL hygromycin B (Pan Biotech, Aidenbach, Germany). M13HS-2 and M13HS-8 hybrid cells: 400 µg/mL G418 (Merck KGaA, Darmstadt, Germany) and 200 µg/mL hygromycin B (Pan Biotech, Aidenbach, Germany). All cells were cultivated in a humidified atmosphere at 37 °C and 5% CO_2_.

### 4.2. Generation of ZEB1-Knock-Out (KO) Cells

The ZEB1-KO variants of the HS578T-Hyg, M13HS-2 and M13HS-8 cells were generated by CRISPR/Cas9. The guide RNA sequence (5′-GAG CAC TTA AGA ATT CAC AG-3′) was obtained from the E-CRISP website (E-CRISP-Version 5.4; https://e-crisp.org/E-CRISP/index.html; accessed on 17 September 2021) and was identical to the published sequence of Kröger and colleagues [19]. Sense and antisense oligonucleotides (gRNA_ZEB1_fwd: 5′-CAC CGA GCA CTT AAG AAT TCA CAG-3′, gRNA_ZEB1_rev: 5′-AAA CCT GTG AAT TCT TAA GTG CTC-3′; Thermo Fisher Scientific, Wesel, Germany) were annealed and ligated into the BbSI digested pX330-U6-Chimeric_BB-CBh-hSpCas9-P2A-PuroR vector-plasmid. This plasmid was constructed by inserting a FseI_p2A-PuroR-bGH Poly(A)_NotI fragment from pcDNA3.1_iCre-T2A-mCherry-p2A-PuroR plasmid into the Fse1/NotI restricted pX330-U6-Chimeric_BB-CBh-hSpCas9 plasmid (pX330-U6-Chimeric_BB-CBh-hSpCas9 was a gift from Feng Zhang; Addgene plasmid #42230; http://n2t.net/addgene:42230 accessed on 1 February 2020; RRID: Addgene_42230). The pX330-U6-Chimeric_BB-CBh-hSpCas9-P2A-PuroR-sgZEB1 plasmid was amplified in DH5α competent bacteria (Thermo Fisher Scientific, Wesel, Germany) and purified using the Nucleospin^®^ Plasmid Transfection-grade kit in accordance with the manufacturer’s instructions (Macherey-Nagel GmbH, Düren, Germany). Cloning was verified by Sanger Sequencing (Eurofins Genomics, Ebersbach, Germany). The sequences were analyzed using SnapGene 5.3.1 software (Dotmatics, Boston, MA, USA).

The cells (HS578T-Hyg, M13HS-2 and M13HS-8) were transfected with pX330-U6-Chimeric_BB-CBh-hSpCas9-P2A-PuroR-sgZEB1 (1 µg) using the jetOPTIMUS^®^ DNA transfection reagent as recommended by the manufacturer (Polyplus, Illkirch, France). To select transfected cells from non-transfected cells, 2 µg/µL puromycin (Thermo Fisher Scientific, Wesel, Germany) was added to the culture medium 24 h after transfection for up to 72 h. Dead cells were removed by washing with phosphate-buffered saline (PBS). Single-cell clones were isolated using 3.2 mm cloning discs (SP Bel-Art/Behr Labor-Technik GmbH, Düsseldorf, Germany) and transferred to 24-well plates (Sarstedt AG and Co. KG, Nümbrecht, Germany) for further propagation. Growing clones were transferred to bigger cell culture flasks once they had reached confluency.

The successful CRISPR/Cas9 ZEB1-KO was validated by Sanger Sequencing. Therefore, genomic DNA was extracted from parental cells and ZEB1-KO variants using the NucleoSpin^®^ Tissue DNA kit as referred to in the instruction manual (Macherey and Nagel, Düren, Germany). The CRISPR/Cas9 edited gene segment was first amplified by PCR (PCR_ZEB1-fwd. 5′-TCC TGT CTT CTA TTC AGG ACC-3′; PCR_ZEB1 rev. 5′-GAA CTT GTT TTC GCG TTT TCC-3′; Thermo Fisher Scientific, Wesel, Germany). Then, the PCR product was analyzed by Sanger sequencing (Eurofins Genomics, Ebersbach, Germany), using the primers Seq_ZEB1_7_fwd. 5′-GGA AAG CAA ACA AGT TAA CCT C-3′ and Seq_ZEB1_7_rev. 5′-TGT AAT CCT TTC ACT CCC TCT C-3′ (Thermo Fisher Scientific, Wesel, Germany). Sequences were analyzed using SnapGene 5.3.1 software (Dotmatics, Bishops Stortford, UK).

Only the successful CRISPR/Cas9 ZEB1-KO variants of the HS578T-Hyg, M13HS-2 and M13HS-8 cells were propagated for further research.

### 4.3. Colony Formation Assay

The cells (5 × 10^2^/per well) were seeded in 6-well plates and were cultivated in complete media for 10 days. Thereafter, the media was removed, cells were washed twice with PBS, were then fixed with 4% paraformaldehyde and stained with 0.5% crystal violet (both reagents were purchased from Merck KGaA, Darmstadt, Germany) for 30 min at room temperature. The plates were thoroughly washed with water and air-dried.

### 4.4. qPCR Analysis of miRNA Expression

Total RNA was isolated from 1.5 × 10^6^ cells using the NucleoSpin^®^ RNA kit in accordance with the instruction manual (Macherey and Nagel, Düren, Germany). cDNA preparation for subsequent miRNA analysis was performed using the TaqMan^®^ advanced miRNA-cDNA synthesis kit as referred to the manufacturer’s instructions (Thermo Fisher Scientific, Wesel, Germany). The following assays were used for miRNA quantification: hsa-miR-200c-3p (assay ID 002300), hsa-mir-34a-5p (assay ID 000426), and hsa-let-7a-5p (assay ID 000377; housekeeping miRNA). The qPCR was run on StepOnePlus™ Real-Time PCR System using cDNA, miRNA assays, and the TagMan^®^ advanced master mix (qPCR cycler and all reagents were from Thermo Fisher Scientific, Wesel, Germany). The qPCR data were analyzed using the StepOne software 2.3 (Thermo Fisher Scientific, Wesel, Germany) and miRNA expression levels were calculated using the ΔC_T_ method, whereby hsa-let-7a-5p was used as an internal control.

### 4.5. Flow Cytometry

CD44 and CD104 expression levels of wildtype and ZEB1-KO cells were determined by flow cytometry using a FACSCalibur flow cytometer (Becton Dickenson, Heidelberg, Germany). Cells were harvested, washed once with PBS, and adjusted to a cell number of 2 × 10^5^ cells/100 µL. Cells were co-stained with either PE and APC matched isotype controls (APC mouse IgG κ; clone 27–35; Becton Dickenson, Heidelberg, Germany; PE mouse IgG2a κ; clone MOPS-173; BioLegend, Amsterdam, The Netherlands) or APC-CD44 and PE-CD104 specific antibodies (APC-CD44; clone G44-26; Becton Dickenson, Heidelberg, Germany; PE-CD104; clone 58XB4; BioLegend, Amsterdam, The Netherlands) for 30 min at 37 °C. Antibody concentrations were used as recommended by the manufacturer’s guidelines. Stained cells were washed once in PBS before flow cytometry analysis. The flow cytometry data were analyzed using the WinMDI 2.8 (https://winmdi.software.informer.com/2.8/ accessed on 1 February 2020).

### 4.6. Western Blot Analysis

The cells were harvested, washed once with PBS, and adjusted to a cell number of 2 × 10^5^ cells/20 µL. Subsequently, 10 µL of 3× Laemmli Sample Buffer was added, and samples were lysed for 10 min at 95 °C. The samples were separated by 10% or 12%, respectively, sodium dodecylsulfate-polyacrylamide gel electrophoresis (SDS-PAGE) and transferred to an Immobilon polyvinyldifluoride (PVDF) membrane (Merck Millipore, Darmstadt, Germany) under semi-dry conditions. The membranes were blocked with 5% bovine serum albumin (BSA) or 5% (*w*/*v*) non-fat milk powder in Tris-buffered saline with 1% (*v*/*v*) Tween 20 (TBS-T). Bands were visualized using the Pierce ECL Western blot substrate (Thermo Fisher Scientific, Wesel, Germany) in accordance with the instruction manual and the Aequoria Macroscopic Imaging System (Hamamatsu Photonics Germany, Herrsching am Ammersee, Germany). Antibodies, which were used in this study are listed in Table 1. The antibody dilutions used in this study were in accordance with the manufacturer’s instructions.

### 4.7. Mammosphere Formation Assay

Mammospheres were generated by seeding cells (500 cells/well) in ultra-low attachment cell 96-well plates (Sarstedt AG and Co KG, Nümbrecht, Germany) in mammosphere formation medium, which is composed of medium I and medium II in a ratio of 1:4. Medium I consists of DMEM/F12 Medium (Pan Biotech, Aidenbach, Germany), 6.6% B27 Supplement (Thermo Fisher Scientific, Wesel, Germany), 20 ng/mL FGF (human recombinant; Merck KGaA, Darmstadt, Germany), 20 ng/mL EGF (human recombinant; Merck KGaA, Darmstadt, Germany) and 0.39 µg/mL hydrocortisone (Sigma-Aldrich, Taufkirchen, Germany). Medium II is composed of Methocult H4100 (Stem Cells Technologies, Cologne, Germany) and DMEM (Pan Biotech, Aidenbach, Germany) in a ratio of 2:3. Mammospheres were cultured for up to 10 days in a humidified atmosphere at 37 °C and 5% CO_2_. The mammosphere formation capacity was determined using the Incucyte^®^ SX5 Live-Cell Analysis System (Sartorius, Göttingen, Germany). Mammospheres with a diameter <60 µm were excluded from analysis.

### 4.8. Confocal Laser Scanning Microscopy

The cell morphology of wildtype and the ZEB1-KO cells was determined by confocal laser scanning microscopy (Leica TCS SP5; Leica Microsystems, Wetzlar, Germany). Cells (2 × 10^4^) were seeded in chamber slides (Nunc Lab-Tek; Thermo Fisher Scientific, Wesel, Germany) for up to 48 h in a humidified atmosphere at 37 °C and 5% CO_2_. Thereafter, cells were fixed with 4% paraformaldehyde (Merck KGaA, Darmstadt, Germany) for 10 min at room temperature and washed with PBS twice. For nuclear staining and visualization of the actin cytoskeleton, cells were first permeabilized with 1% Triton X-100 ((*v*/*v*) in PBS) for 5 min at room temperature and then washed with PBS twice. The actin cytoskeleton was stained by Alexa Fluor^®^ 568-phalloidin (Thermo Fisher Scientific, Wesel, Germany) for 30 min at room temperature in the dark. Nuclear staining was performed with SYTOX^®^-Green (Thermo Fisher Scientific, Wesel, Germany) for 15 min at room temperature in the dark. Cells were thoroughly washed with PBS and mounted with Fluoromount (Thermo Fisher Scientific, Wesel, Germany).

### 4.9. Scratch/Wound-Healing Assay

Cells (2 × 10^5^) were seeded in triplicates for 24 h in a 24-well plate (Sarstedt AG and Co KG, Nümbrecht, Germany), which was sufficient for reaching 100% confluency. Then, a scratch/wound was set using a 100 µL pipette tip. Cell debris was removed by washing once with PBS. Fresh media (1.5 mL) was applied and closing of the scratch/wound was recorded using an Incucyte^®^ SX5 LiveCell Imaging system (Sartorius Lab Instruments GmbH, Göttingen, Germany), whereby transmission light images were taken every four hours for a total time of 24 h at 37 °C and 5% CO_2_. The migration area was analyzed using Fiji software 1.54f (Image J; https://Fiji.sc accessed on 1 February 2020). The scratch/wound size at t = 12 h was calculated in relation to the scratch/wound size at t = 0 h, which was set to 100%.

### 4.10. Transwell/Boydenchamber Assay and Invasion Assay

For a normal transwell migration assay, the cells (2 × 10^5^) were seeded in the upper chamber of a transwell insert (diameter 12 mm, 8 µm pore size; Becton Dickenson, Heidelberg, Germany) of a 12-well plate (Sarstedt AG and Co KG, Nümbrecht, Germany) in 400 µL serum-free media. The lower compartment was filled with 500 µL complete media.

For an invasion assay, transwell inserts were coated with Geltrex as recommended by the manufacturer’s instructions. Briefly, 100 µL of a 1:4 dilution of 4 °C cold Geltrex with 4 °C cold PBS was filled in transwell inserts (diameter 6.5 mm, 8 µm pore size; Becton Dickenson, Heidelberg, Germany) and allowed to polymerize for 60 min at 37 °C. Subsequently, cells (6.5 × 10^4^) in serum-free media were seeded on top of the polymerized Geltrex matrix. The lower compartment was filled with 250 µL complete media.

Cells were cultivated for 24 h at 37 °C and 5% CO_2_. Subsequently, the remaining cells in the upper compartment were carefully removed with a cotton swab. Cells in the lower compartment were fixed with 4% paraformaldehyde (Agilent Technologies Deutschland GmbH, Waldbronn, Germany) for 15 min at room temperature. Fixed cells were washed twice with PBS and then stained with 1% crystal violet staining solution (Merck KGaA, Darmstadt, Germany) for 30 min at room temperature. Stained cells were thoroughly washed in water and air-dried. Images were taken with an inverted microscope (Leica DM IRB; Leica, Wetzlar, Germany) and the Zeiss Labscope software 3.4.2 (Carl Zeiss Microscopy GmbH, Jena, Germany). In each experiment, six randomly chosen images were taken. The images were analyzed using Fiji software 1.54f (Image J; https://Fiji.sc, accessed on 1 February 2020).

### 4.11. AldeRed Assay

The AldeRed aldehyde dehydrogenase 1 (ALDH1) detection assay was performed as recommended in the manufacturer’s instructions and described previously [52]. Briefly, 2 × 10^5^ cells were used for one measurement, whereby half of the cell suspension was used for control purposes (treatment of cells with the ALDH1 inhibitor diethylaminobenzaldehyde (DEAB)). The cells were incubated for 30 min at 37 °C in the dark, washed once, and resuspended in 500 µL AldeRed assay buffer. All the samples were stored on ice before the flow cytometry analysis (FACSCalibur; Becton Dickenson, Heidelberg, Germany). The flow cytometry data were analyzed using WinMDI 2.8 (https://winmdi.software.informer.com/2.8/, accessed on 1 February 2020).

### 4.12. Statistical Analysis

A statistical analysis was performed using the GraphPad PRISM software 8.4.2 (https://www.graphpad.com, accessed on 1 February 2020). A detailed description of which statistical test was used is given in the appropriate figure legends.

## 5. Conclusion

This study aimed to clarify the role of ZEB1 in M13HS tumor hybrids that were derived from spontaneous fusion events between the M13SV1-EGFP-Neo human breast epithelial cells and HS578T-Hyg human breast cancer cells [51]. Thereby, particular attention was drawn to a prospective mixed/hybrid E/M phenotype and CSC properties due to SNAIL and ZEB1 co-expression of M13HS tumor hybrids and EMT-related properties. However, the data presented here indicates that the ZEB1-associated CSC characteristics and EMT-related properties might be more complex. Even though ZEB1 was successfully and stably knocked-out in HS578T-Hyg breast cancer cells and M13HS tumor hybrids, the phenotype of the ZEB1-KO cells was only moderately altered as compared to their wildtype counterparts. As indicated, neither a CDH1 upregulation nor increased miR-200c-3p levels were observed. Similarly, a markedly higher ALDH1 activity was observed in M13HS-2 ZEB1-KO cells. Whether this was attributed to the up-regulation of other EMT-related transcription factor markers, such as SNAIL and/or SLUG and/or ZEB2, which might replace ZEB1 function in the ZEB1-KO cells, or to other, not yet identified CRISPR/Cas9-related processes or cell fusion-related mechanisms, remains to be elucidated in future work. In summary, although the ZEB1 knock-out resulted in only moderate phenotypic changes, our data support the role of ZEB1 in EMT and stemness.

## Figures and Tables

**Figure 1 ijms-24-17310-f001:**
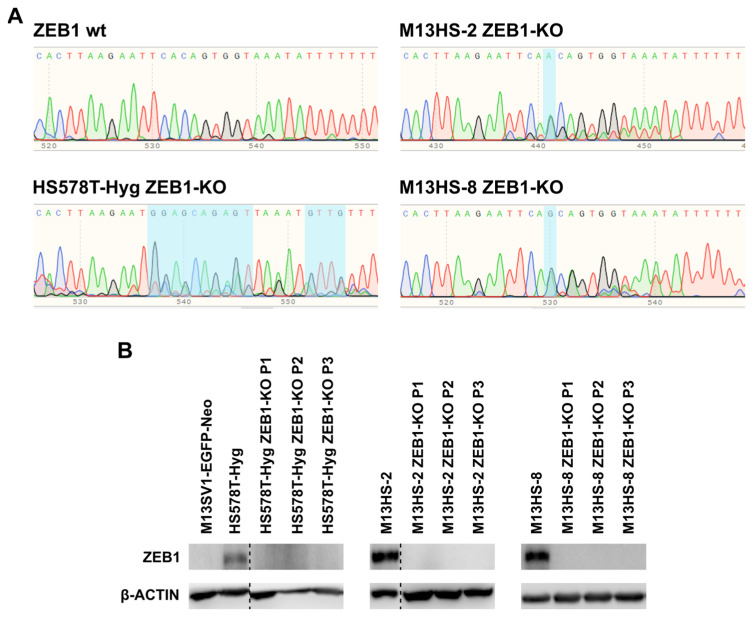
ZEB1 was successfully knocked-out in the HS578T-Hyg breast cancer cells and M13HS-2 and -8 tumor hybrids. (**A**) Sequencing data indicate a nucleotide insertion in the M13HS-2 and -8 tumor hybrids, whereas, in the HS578T-Hyg breast cancer cells, a more chaotic genome editing was observed. Alterations are marked in light blue. (**B**) Representative ZEB1 Western blot data of three consecutive passages (P1, P2, and P3). ZEB1 blots for the HS578T-Hyg/HS578T ZEB1-KO and M13HS-2/M13HS-2 ZEB1-KO cells were derived from one blot, where bands from other samples were cut off (indicated by a dashed line). P1, P2, and P3 indicate three consecutive passage numbers. Original Western blot data are shown in Appendix A.

**Figure 2 ijms-24-17310-f002:**
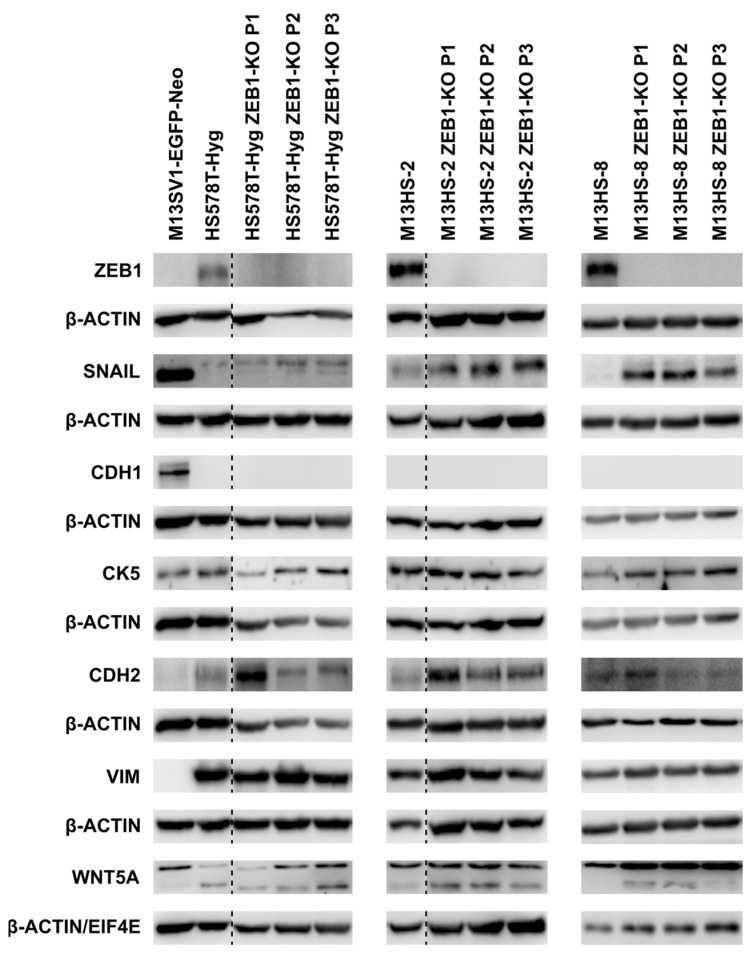
The protein expression pattern of the ZEB1-KO cells was weakly altered in comparison to wildtype cells. Shown are representative Western blot data of at least three independent experiments and three consecutive passages (P1, P2, and P3). Note that some blots, such as CDH2 and CK5 of the HS578T-Hyg cells, CDH1 and CK5 of M13HS-2 and -8 cells were double stained and thus have identical β-actin housekeeping bands. EIF4E was used as a housekeeping gene for WNT5A expression of the M13HS-8 and M13HS-8 ZEB1-KO cells. Blots for the HS578T-Hyg/HS578T ZEB1-KO and M13HS-2/M13HS-2 ZEB1-KO cells were derived from one blot. Here, bands from other samples were cut off (indicated by a dashed line). The arrow marks the active form of WNT5A. P1, P2, and P3 indicate three consecutive passage numbers. Original Western blot data are shown in Appendix A.

**Figure 3 ijms-24-17310-f003:**
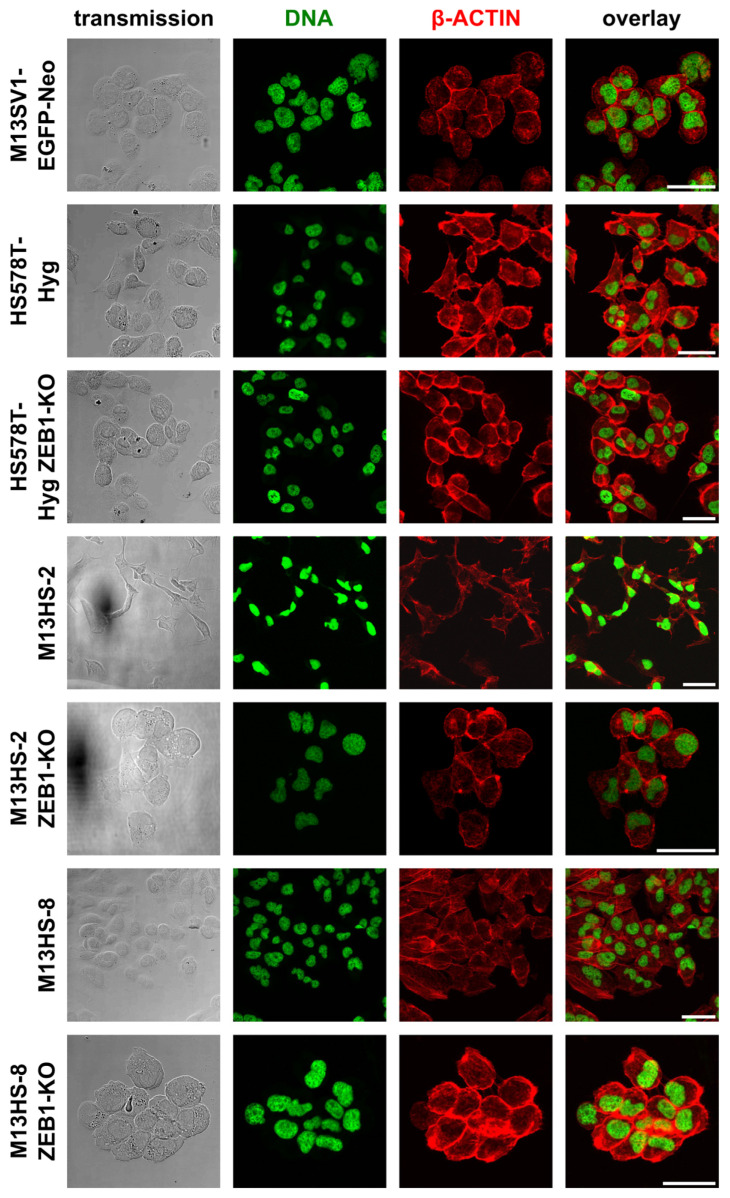
The ZEB1-KO cells exhibit a round-shaped epithelial morphology. Cells were seeded on chamber slides, fixed and stained for DNA and β-ACTIN. Shown are representative data of two independent measurements. Bar = 50 µm.

**Figure 4 ijms-24-17310-f004:**
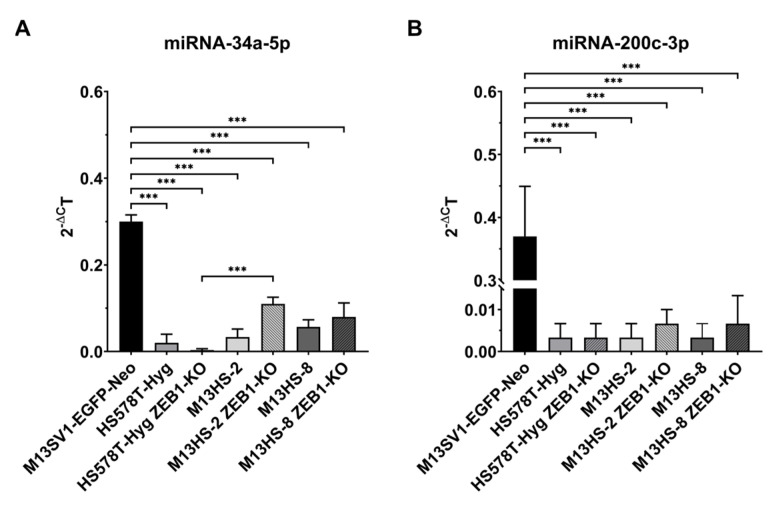
miRNA expression pattern of the ZEB1-KO cells and wildtype cells. (**A**) miRNA-34a-5p data. (**B**) miRNA-200c-3p data. Shown are the mean ± S.E.M of three independent experiments. Statistical significance was calculated using a one-way ANOVA and Tukey post-hoc test. *** = *p* < 0.001.

**Figure 5 ijms-24-17310-f005:**
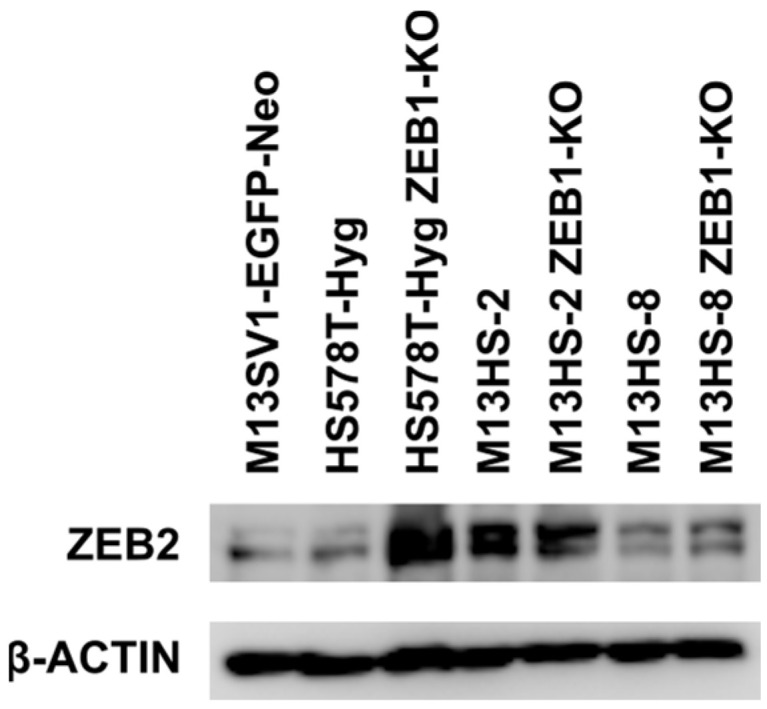
ZEB2 expression of ZEB1-KO cells and wildtype cells. Slightly higher ZEB2 levels were observed in HS578T-Hyg ZEB1-KO cells, whereas the M13HS-2 and M13HS-2 ZEB1-KO cells, the M13HS-8 and M13HS-8 ZEB1-KO cells, respectively, possessed comparable ZEB2 expression levels. Shown are representative data from two independent experiments.

**Figure 6 ijms-24-17310-f006:**
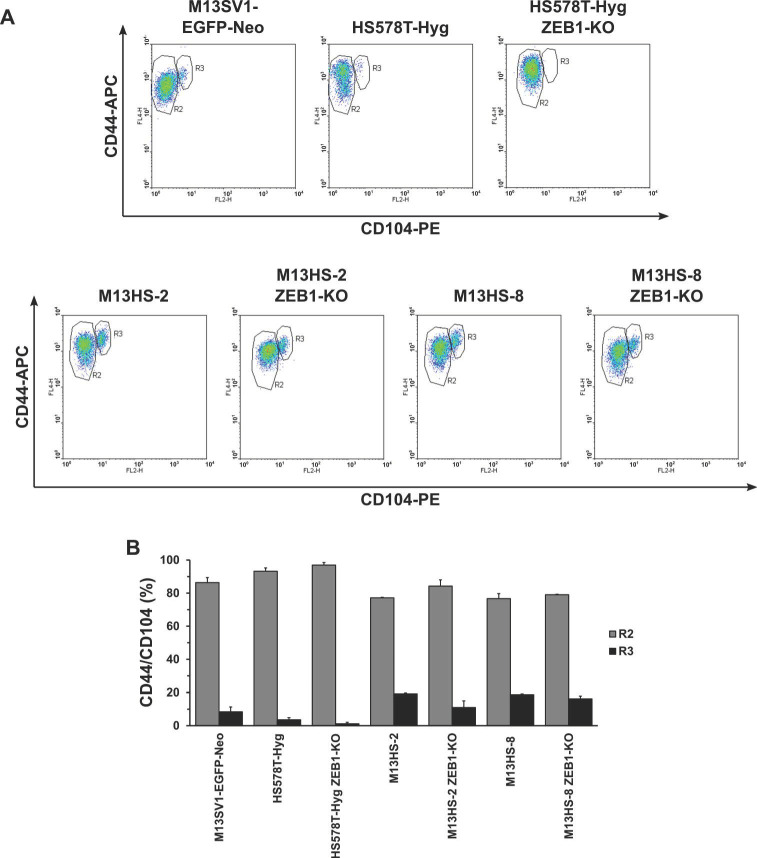
CD44/CD104 expression pattern of ZEB1-KO cells and wildtype cells. (**A**) Representative FACS data of two independent experiments (mean ± STD). (**B**) The quantification of flow cytometry data indicates that M13HS tumor hybrids harbor a higher fraction of population 2 cells (R3). Similarly, ZEB1-KO was correlated with a reduction of R3 population cells.

**Figure 7 ijms-24-17310-f007:**
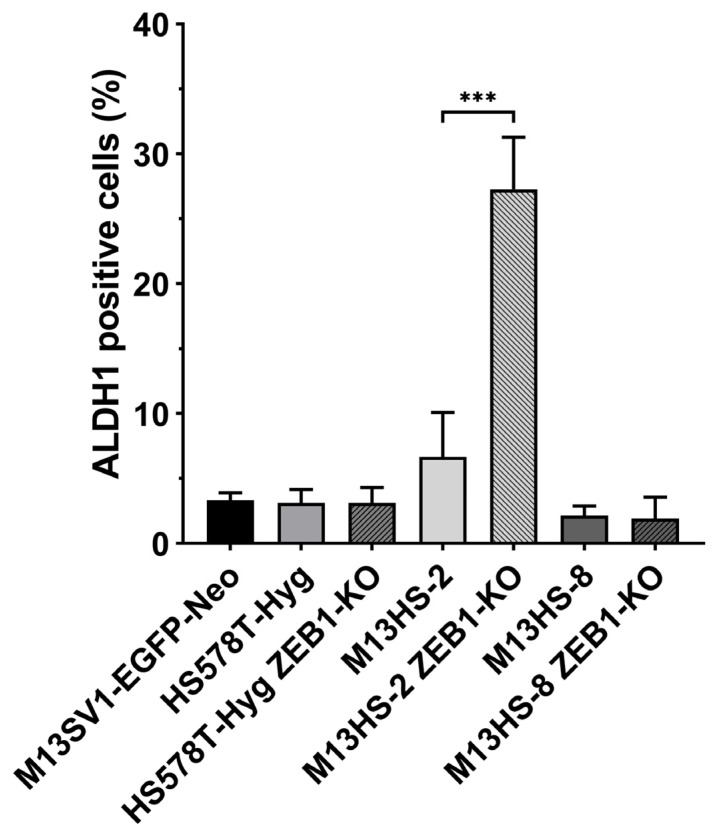
M13HS-2 ZEB1-KO cells are enriched in ALDH1-positive cells, as determined by flow cytometry. Shown are the mean ± S.E.M. of ALDH1-positive cells (ALDH data minus DEAB control data) of at least four independent experiments. Statistical significance was calculated using a one-way ANOVA and Tukey post-hoc test. *** = *p* < 0.001.

**Figure 8 ijms-24-17310-f008:**
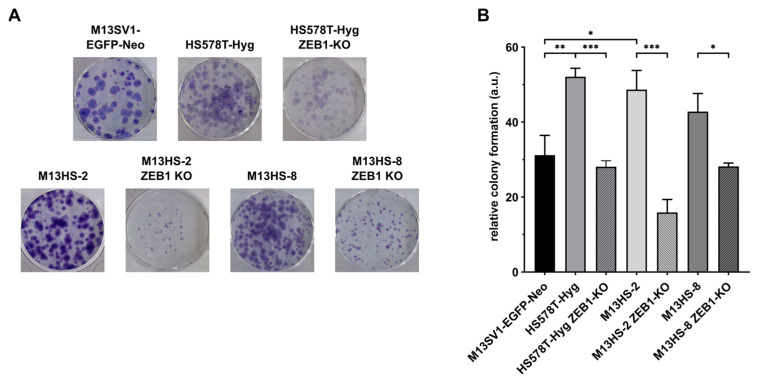
ZEB1-KO cells exhibit a significantly decreased colony formation capacity. (**A**) Representative images of colony formation assays. (**B**) Shown are the mean ± S.E.M of three independent experiments. Statistical significance was calculated using a one-way ANOVA and Tukey post-hoc test. * = *p* < 0.05, ** = *p* < 0.01, *** = *p* < 0.001.

**Figure 9 ijms-24-17310-f009:**
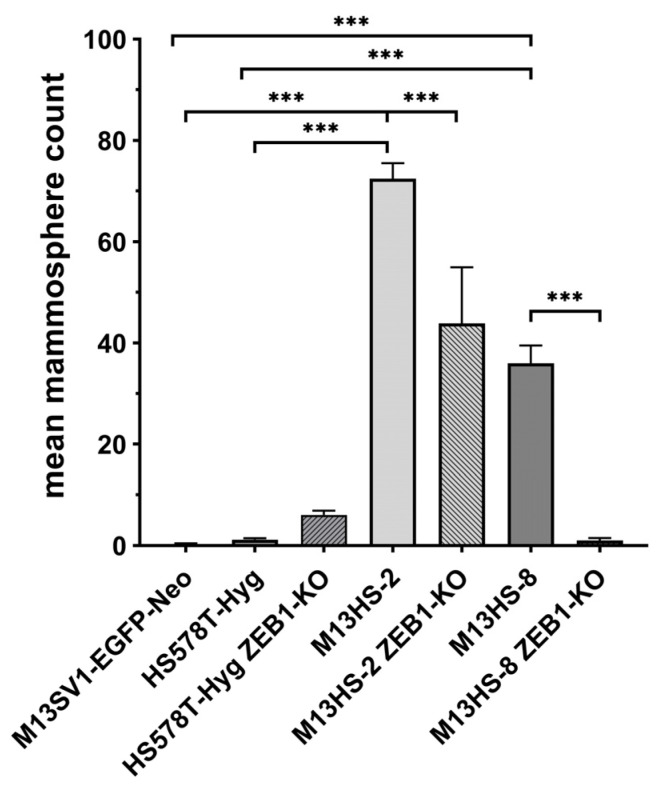
ZEB1-KO cells exhibit an altered mammosphere formation capacity. Shown are the mean ± S.E.M of three independent experiments. Statistical significance was calculated using a one-way ANOVA and Tukey’s post-hoc test. *** = *p* < 0.001.

**Figure 10 ijms-24-17310-f010:**
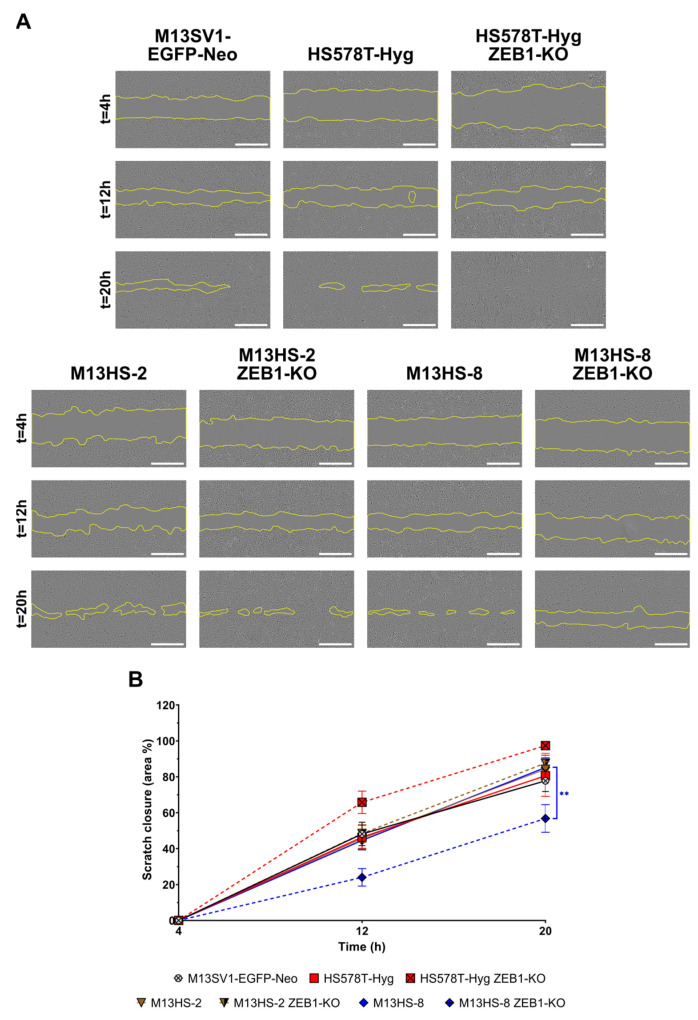
ZEB1-KO cells exhibit unique migration properties. Shown are representative images (**A**) and the mean ± S.E.M of three independent experiments (**B**). Statistical significance was calculated using a two-way ANOVA and Tukey’s post-hoc test. ** = *p* < 0.01. Bar = 100 µm.

**Figure 11 ijms-24-17310-f011:**
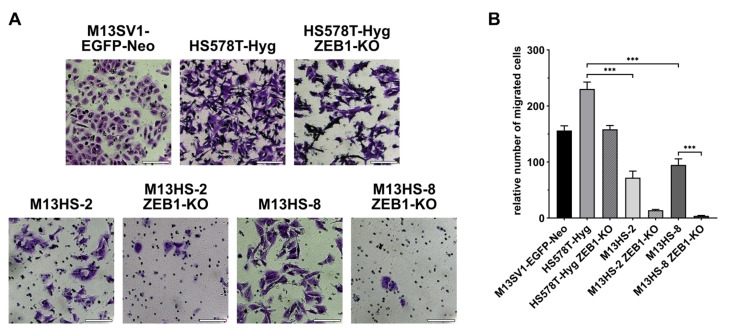
ZEB1-KO cells were less migratory active in Transwell/Boyden chamber migration assays than the wildtype cells. Shown are representative images (**A**) and the mean ± S.E.M of three independent experiments (**B**). Statistical significance was calculated using a Kruskal–Wallis non-parametric test and Dunn’s post-hoc test. *** = *p* < 0.001. Bar = 100 µm.

**Figure 12 ijms-24-17310-f012:**
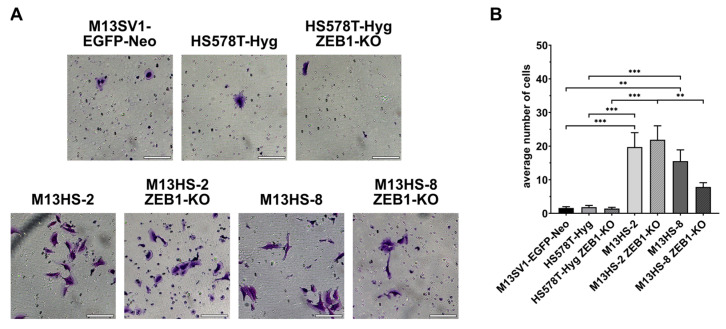
ZEB1-KO cells exhibit different invasion capacities. Shown are representative images (**A**) and the mean ± S.E.M of three independent experiments (**B**). Statistical significance was calculated using a one-way ANOVA and Tukey’s post-hoc test. ** = *p* < 0.01; *** = *p* < 0.001. Bar = 100 µm.

**Table 1 ijms-24-17310-t001:** Used antibodies of this study.

Antibody	Clone; Catalog Number	Manufacturer
anti-β-ACTIN (mouse monoclonal)	clone AC-15; A5441	Merck KGaA, Darmstadt, Germany
E-CADHERIN (CDH1) (rabbit monoclonal)	clone 24E10; 3195S	Cell Signaling Technology Europe B.V., Frankfurt am Main, Germany
eIF4E (rabbit polyclonal)	9742S	Cell Signaling Technology Europe B.V., Frankfurt am Main, Germany
CYTOKERATIN-5 (rabbit monoclonal)	clone D4U8Q; 25807S	Cell Signaling Technology Europe B.V., Frankfurt am Main, Germany
N-CADHERIN (CDH2) (mouse monoclonal)	clone 13A9; 14215S	Cell Signaling Technology Europe B.V., Frankfurt am Main, Germany
SNAIL (rabbit monoclonal)	clone C15D3; 3879S	Cell Signaling Technology Europe B.V., Frankfurt am Main, Germany
VIMENTIN (rabbit polyclonal)	3932S	Cell Signaling Technology Europe B.V., Frankfurt am Main, Germany
WNT5A (rabbit monoclonal)	clone G.307.7; MA5-14946	Thermo Fisher Scientific, Wesel, Germany
ZEB1 (rabbit monoclonal)	clone D808D3; 3396S	Cell Signaling Technology Europe B.V., Frankfurt am Main, Germany
ZEB2 (rabbit polyclonal)	14026-1-AP	Proteintech Germany GmbH Planegg-Martinsried, Germany
anti-mouse IgG, HRP linked (horse polyclonal)	7076S	Cell Signaling Technology Europe B.V., Frankfurt am Main, Germany
anti-rabbit IgG, HRP linked (horse polyclonal)	7074S	Cell Signaling Technology Europe B.V., Frankfurt am Main, Germany

## Data Availability

All data will be shared upon reasonable request.

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
