# Peer review of "Altered Phenotypes of Breast Epithelial × Breast Cancer Hybrids after ZEB1 Knock-Out"

_ijms, 2023, doi:10.3390/ijms242417310_

Round 1
Reviewer 1 Report
Comments and Suggestions for Authors
In the manuscript titled "Altered phenotypes of breast epithelial × breast cancer hybrids two after ZEB1 knock-out", the knock-out of the ZEB1 gene with CRISPR/Cas9 in fused M13HS tumor cells showing different character behavior and the effect of this related gene knock-out on the EMT behavior of the cells were examined. The studies carried out in the manuscript are well explained in the materials and methods section. Generally, the manuscript's handling of the subject and its experimental planning have been well evaluated. However, some improvements need to be made in the manuscript. These are;
1. In the manuscript, the ZEB1 gene was knocked out with the CRISPR/Cas9 system. There are studies in which the ZEB1 gene was knocked out with the CRISPR/Cas9 system in different cancer cell lines. These studies also examined the effects of ZEB1 knock-out on EMT. The effect of ZEB1 knock-out in these studies should be discussed in the introduction. Some examples of these studies are:
Al-Sammarraie N, Ray SK. Applications of CRISPR-Cas9 Technology to Genome Editing in Glioblastoma Multiforme. Cells. 2021 Sep 7;10(9):2342. doi: 10.3390/cells10092342. PMID: 34571991; PMCID: PMC8468137.
Jägle S, Dertmann A, Schrempp M, Hecht A. ZEB1 is neither sufficient nor required for epithelial-mesenchymal transition in LS174T colorectal cancer cells. Biochem Biophys Res Commun. 2017 Jan 22;482(4):1226-1232. doi: 10.1016/j.bbrc.2016.12.017. Epub 2016 Dec 5. PMID: 27923654.
2. In the explanation of Figure 1B, it would be helpful to explain that P1, P2, and P3 indicate the passage number of the cells.
3. The guide RNA with the sequence 5'-GAGCACTTAAGAATTCACAG-3' used in the study targets 19 additional genes with three nucleotide mismatches. I advise authors to use gRNAs with fewer mismatches in their following studies or use Cas9 enzymes such as eCas9 with more target specificity. In this way, gRNAs with many off-target can affect the study results.
4. It would be better to include the ZEB2 gene, which is a paralogue of the ZEB1 gene, in the study. The increase in SNAIL expression in cells with the ZEB1 gene knocked out may be due to the increasing expression of the ZEB2 gene. In this way, the decrease in ZEB1 expression in cells may be compensated by the increase in ZEB2 expression. If there are studies on this subject, discuss them in the conclusion section of the manuscript.
Author Response
Reviewer #1
We would like to thank Reviewer#1 for taking the time to read the manuscript critically and for providing us helpful comments to improve the quality of the manuscript. In the following, we will address his/her comments point by point. Changes in the revised manuscript are marked by a red font color.
In the manuscript titled "Altered phenotypes of breast epithelial × breast cancer hybrids two after ZEB1 knock-out", the knock-out of the ZEB1 gene with CRISPR/Cas9 in fused M13HS tumor cells showing different character behavior and the effect of this related gene knock-out on the EMT behavior of the cells were examined. The studies carried out in the manuscript are well explained in the materials and methods section. Generally, the manuscript's handling of the subject and its experimental planning have been well evaluated. However, some improvements need to be made in the manuscript. These are;
- In the manuscript, the ZEB1 gene was knocked out with the CRISPR/Cas9 system. There are studies in which the ZEB1 gene was knocked out with the CRISPR/Cas9 system in different cancer cell lines. These studies also examined the effects of ZEB1 knock-out on EMT. The effect of ZEB1 knock-out in these studies should be discussed in the introduction. Some examples of these studies are:
Al-Sammarraie N, Ray SK. Applications of CRISPR-Cas9 Technology to Genome Editing in Glioblastoma Multiforme. Cells. 2021 Sep 7;10(9):2342. doi: 10.3390/cells10092342. PMID: 34571991; PMCID: PMC8468137.
Jägle S, Dertmann A, Schrempp M, Hecht A. ZEB1 is neither sufficient nor required for epithelial-mesenchymal transition in LS174T colorectal cancer cells. Biochem Biophys Res Commun. 2017 Jan 22;482(4):1226-1232. doi: 10.1016/j.bbrc.2016.12.017. Epub 2016 Dec 5. PMID: 27923654.
We would like to thank the reviewer for his/her helpful comment. The publication by Jägle et al. was included in the "Introduction and discussion" section. The publication by Al-Sammarraie et al. was not included as it is a review paper on CRISPR/Cas9 in genome editing in GBM. Instead, citation #76 of this review was included in the revised manuscript (Introduction) due to CRISPR/Cas9-mediated ZEB1-KO in glioblastoma cells. In addition, a recent paper by Sanchez-Tillo and colleagues was added in which they described opposing functions of ZEB1 in KRAS- and BRAF-mutated colorectal carcinomas (Sanchez-Tillo et al.; The EMT factor ZEB1 paradoxically inhibits EMT in BRAF-mutant carcinomas. JCI Insight. 2023 Oct 23;8(20):e164629. doi: 10.1172/jci.insight.164629.).
- In the explanation of Figure 1B, it would be helpful to explain that P1, P2, and P3 indicate the passage number of the cells.
We would like to thank the reviewer for this improving comment. This was added to the legends of Figure 1 and Figure 2.
- The guide RNA with the sequence 5'-GAGCACTTAAGAATTCACAG-3' used in the study targets 19 additional genes with three nucleotide mismatches. I advise authors to use gRNAs with fewer mismatches in their following studies or use Cas9 enzymes such as eCas9 with more target specificity. In this way, gRNAs with many off-target can affect the study results.
We agree with the reviewer and thank him/her for this helpful comment. We used the E-CRISP website to search for suitable ZEB1-specific gRNA sequences. The first three gRNA sequences with the best SAE score were selected for CRISPR7Cas9-KO experiments (ZEB1_17_135107; GG TAA CAC TGT CTG GTC TGT NGG; ZEB1_45_173709; GC CTC TAT CAC AAT ATG GAC NGG; ZEB1_7_193010; GA GCA CTT AAG AAT TCA CAG NGG). Interestingly, the third gRNA sequence was identical to the gRNA sequence of Kroger et al. (PNAS USA 2019, 116, (15), 7353-7362). All three gRNA sequences were tested, but a complete ZEB1-KO was only achieved with the third gRNA sequence (ZEB1_7_193010), which was also used in the work of Kroger et al.
- It would be better to include the ZEB2 gene, which is a paralogue of the ZEB1 gene, in the study. The increase in SNAIL expression in cells with the ZEB1 gene knocked out may be due to the increasing expression of the ZEB2 gene. In this way, the decrease in ZEB1 expression in cells may be compensated by the increase in ZEB2 expression. If there are studies on this subject, discuss them in the conclusion section of the manuscript.
Thank you for this suggestion. We performed ZEB2 western blots and found that ZEB2 was expressed in all cells. Interestingly, significantly higher ZEB2 expression was observed in HS578T-Hyg-ZEB1-KO cells than in HS578T-Hyg wild-type cells. In contrast, M13HS tumor hybrids and their ZEB1-KO variants exhibited comparable ZEB2 expression levels. Thus, the ZEB2 expression of ZEB1-KO cells could be an explanation for the still low miRNA-200c-3p levels in these cells. However, ZEB2 expression was also observed in M13SV1-EGFP neo-crustal epithelial cells expressing high levels of miRNA-200c-3p. Therefore, the possible influence of ZEB2 on miRNA-200c-3p expression should be further investigated in ongoing studies.

Reviewer 2 Report
Comments and Suggestions for Authors
It is very interesting manuscript, but introduction should be more kinder to the readers. The authors used M13HS2 and 8 among dozen of Hybrid reported Ref 26 and 27. The readers would be bewildered to see this system without background. Please state history of their research and continued interest in the first paragraph of the introduction.
Wy they picked up these clones among others. After that the story would be fine. And state the merit of this system (hybrid cells). Spontaneous fusion may effect the following events OR just the authors just took advantage of the phenotype obtained by serendipity?
E-cadherin may called as CDH1 instead of E-CADH.
Author Response
Reviewer #2
We would like to thank Reviewer#2 for taking the time to read the manuscript critically and for providing us helpful comments to improve the quality of the manuscript. In the following, we will address his/her comments point by point. Changes in the revised manuscript are marked by a red font color.
It is very interesting manuscript, but introduction should be more kinder to the readers. The authors used M13HS2 and 8 among dozen of Hybrid reported Ref 26 and 27. The readers would be bewildered to see this system without background. Please state history of their research and continued interest in the first paragraph of the introduction.
Wy they picked up these clones among others. After that the story would be fine. And state the merit of this system (hybrid cells). Spontaneous fusion may effect the following events OR just the authors just took advantage of the phenotype obtained by serendipity?
We would like to thank the reviewer for his/her helpful comment. A short passage about cell-cell fusion and why we used these two tumor hybrid clones in this study was included in the introduction.
E-cadherin may called as CDH1 instead of E-CADH.
Thank you for this comment. E-CADH has been replaced by CDH1.
